# Returning to Leisure Activity Post-Stroke: Barriers and Facilitators to Engagement

**DOI:** 10.3390/ijerph192114587

**Published:** 2022-11-07

**Authors:** Joanna Harrison, Clare Thetford, Matthew J. Reeves, Christopher Brown, Miland Joshi, Caroline Watkins

**Affiliations:** 1Synthesis, Economic Evaluation and Decision Science (SEEDS), Applied Health Research hub (AHRh), University of Central Lancashire, Preston PR1 2HE, UK; 2Stroke Research Team, Faculty of Health and Care, University of Central Lancashire, Preston PR1 2HE, UK; 3UCLan Research Centre for Sport, Physical Activity & Performance, Faculty of Allied Health and Wellbeing, University of Central Lancashire, Preston PR1 2HE, UK; 4Lancashire Clinical Trials Unit, University of Central Lancashire, Preston PR1 2HE, UK; 5Lancashire Institute for Global Health and Well-being, University of Central Lancashire, Preston PR1 2HE, UK

**Keywords:** stroke, TIA, leisure activity, engagement, qualitative

## Abstract

Objectives: To identify barriers and facilitators to engagement when returning to, or participating in, leisure activity post-stroke or Transient Ischemic Attack (TIA). Design: Sequential explanatory, mixed methods study. Setting: 21 hospital sites across England, Wales and Northern Ireland. Participants: Adults with a clinical diagnosis of first/recurrent stroke or TIA. Patients approaching end of life were excluded. Participants were recruited as in-patients or at first clinic appointment and a baseline questionnaire was completed. A 6-month follow-up questionnaire was sent to participants for self-completion. Open-text questions were asked about barriers and facilitators when returning to, or participating in, leisure activity. Responses were thematically analysed and explored by participant characteristics, including type of leisure activity undertaken. Characteristics also included measures of socioeconomic deprivation, mood, fatigue and disability. Results: 2000 participants returned a 6-month follow-up questionnaire (78% stroke, 22% TIA); 1045 participants responded to a question on barriers and 820 on facilitators. Twelve themes were identified and the proportion of responses were reported (%). Barriers: physical difficulties (69%), lower energy levels (17%), loss of independence (11%), psychological difficulties (10%), hidden disabilities (7%), and delay or lack of healthcare provision (3%). Facilitators: family support (35%), healthcare support (27%), well-being and fitness (22%), friendship support (20%), self-management (19%), and returning to normality (9%). ‘Physical difficulties’ was the most reported barrier across all participant characteristics and activity types. Family support was the most reported facilitator except for those with greater disability, where it was healthcare support and those without fatigue where it was well-being and exercise. Conclusions: Physical difficulties and lack of energy are problematic for stroke and TIA survivors who want to return to or participate in leisure activity. Healthcare support alone cannot overcome all practical and emotional issues related to leisure activity engagement. Family support and improving well-being are important facilitators and future research should explore these mechanisms further.

## 1. Introduction

Stroke is a worldwide problem that causes significant mortality and disability [1,2]. Between 2015 and 2035, the annual number of strokes in the UK may increase by 60% and the number of stroke survivors by 120% [3]. Long-term survival after stroke has gradually improved in recent decades, with 28% of those aged under 65 surviving at least 15 years post-stroke [4]. As the stroke population lives longer, it is imperative to address their physical, psychological, and social needs for rehabilitation and recovery.

Physical activity is a key element of stroke rehabilitation and is associated with a reduction in cardiovascular risk factors, moderating the risk of recurrent strokes and mortality [5,6]. However, physical activity may not always be feasible, and other forms of leisure or social activity may be more appropriate. A review of leisure therapy in stroke rehabilitation suggested that recreational activities contribute to short-term improvements in psychological outcomes, including quality of life and mood [7]. Conversely, a decrease in leisure activity participation was associated with an increase in depressive symptoms at one-year post-stroke [8]. Leisure engagement in later life (including activities such as reading books and gardening), has also been associated with enhanced survival among older adults, regardless of their earlier participation habits and health [9].

Engagement in leisure activity post-stroke is not without issue; frequency of social and leisure participation decreases over two-years post-stroke, with older adults, those with a lower education level and dependence in activities of daily living, all factors for less favourable participation outcomes [10]. Moreover, two thirds of stroke survivors between one to five years post-stroke, experience a change for the worse in number or type of leisure activities or interests [11,12]. For mild-stroke survivors, where motor function is minimally compromised, almost 30% are unable to participate in some life habits (including leisure activity) without difficulty at one-year post-stroke [13]. Post-Transient Ischemic Attack (TIA), residual cognitive and physical impairments such as anxiety and fatigue have also been reported to impact on participation in social activities and close relationships [14]. Long-term participation in social and leisure activity post-stroke is also influenced by a range of personal and contextual factors that change with time, including personal characteristics, and having the motivation and capacity to participate [15].

We have previously shown that engagement in leisure activities by six-months post-stroke or TIA, reduces on average by 22% compared to pre-event levels and notably so for older adults, females, and those living in higher socioeconomic deprivation [16]. To further our understanding of what helps or hinders engagement in leisure activity post-stroke or TIA, we used existing data from this study to undertake a qualitative analysis. We aimed to explore the barriers and facilitators to engagement when returning to, or participating in, leisure activity post-stroke or TIA. Secondary aims included exploration of the barriers and facilitators by participant characteristics including demographics, clinical factors and type of important activity undertaken.

## 2. Methods

### 2.1. Study Design 

A self-report questionnaire on leisure activity was administered to participants who had a diagnosis of stroke or TIA at baseline and at 6-month follow-up. Full details of the study have been reported elsewhere [16]. Using a sequential, explanatory, mixed-methodology [17], qualitative data were identified from the 6-month follow-up questionnaire, analysed and were used to contextualise the findings from the prior study.

Setting: This study was conducted across 21 hospital sites, geographically spread across England, Wales and Northern Ireland.

Participants: Eligible participants were: (1) adults aged 18 or over; (2) those with a clinical diagnosis of new first or recurrent stroke or TIA; (3) a pre-stroke modified Rankin Scale (mRS) score of ≤3 (An mRS score of 0 = no symptoms; 1 = no significant disability despite symptoms and ability to perform all usual activities; 2 = slight disability and an inability to perform all previous activities; and 3 = moderate disability where some assistance with activities is required but are able to walk without assistance. mRS scores 4–5 suggest it would be difficult for individuals to engage in many leisure activities due to the severity of their stroke/TIA-related disability); (4) the capacity to give written informed consent, or a suitable consultee able to provide consent; and (5) could communicate in English or had a suitable consultee who could assist in questionnaire completion. Possible participants were excluded if their clinical care team identified them as being near the end of life.

Sampling and Sample Size: Participating sites were asked to recruit the first 15 people that consented to participation for each month of recruitment (December 2017–April 2019). A 2:1 ratio for stroke to TIA participants was encouraged to ensure the study focused on leisure after stroke, without ignoring residual difficulties post-TIA.

Procedure: Participants were recruited as an in-patient or at a first post-stroke/TIA clinic appointment by a Stroke Research Nurse. Following consent, participants completed a baseline questionnaire. Follow-up questionnaires were posted to participants by the Lancashire Clinical Trials Unit (LCTU), 6-months post-stroke or TIA. Both paper questionnaires were self-completed (or with the assistance of a suitable consultee). Completed questionnaires were returned to LCTU and were entered into Stata 15/16.

### 2.2. Questionnaires

The 6-month questionnaire included two open-text questions regarding perceived barriers and facilitators to engaging in leisure activities: ‘Has anything made it more difficult for you get back to doing the activities that you enjoy?’ and ‘Has anything helped you get back to doing the activities that you enjoy?’. A further open-text question asked ‘what leisure activities are most important to you now?’. Participant data included event type, age, sex, ethnicity, impairments (visual, hearing, and speech), social circumstances, leisure participation, and socioeconomic deprivation calculated using the Index of Multiple Deprivation (IMD) for England. In our model, IMD was treated as a scale of 1 to 5 quintiles (higher values indicate lower socioeconomic deprivation and, hence, a higher socioeconomic status). The questionnaire also included responses about fatigue post-stroke (yes/no response), mood using the Yale-Brown Single item screening question for depression [18] and self-assessed disability using the mRS [19]. Scores from the mRS were reported using the following categories (0–1) no symptoms to no significant disability, (2–3) slight to moderate disability and (4–5) moderate severe to severe disability. 

### 2.3. Data Analysis

Questionnaire data from the 6-month follow-up were also entered into a Microsoft (MS) Excel spreadsheet. The open-text responses regarding barriers and facilitators were coded by a researcher (JH) using a constant comparative approach [20]. Using a coding frame, codes were categorized into conceptual themes for the two sets of responses. The coding frame was subsequently explored by the wider research team and a consensus of understanding was reached. Subsequent themes related to physical, social or emotional phenomena and represented the substantial meaning of the response, for example, ‘the support of my daughter and grandchildren’ was coded as family support.

Participant characteristics (demographics and clinical factors) for those who had responded to the two questions were organised according to the attributed theme (separate spreadsheet for each theme). Where a participant gave a response that included multiple themes, their characteristics were included in each relevant theme. Within each theme spreadsheet, the distribution of participant characteristics was identified and reported (Appendix A). For example, we identified the number of female participants that indicated family support was helpful to them. This number was then compared to the number of females who had answered the ‘has anything helped’ question (facilitator) and a proportion calculated (%). The three highest proportions of participant characteristic within each of the barrier and facilitator themes were subsequently reported.

The responses to ‘what leisure activities are most important to you now?’ were categorised by a researcher (JH) to reflect the main types of activities identified: fitness and well-being (e.g., going to the gym, mindfulness), every-day (reading, watching TV), craft and hobby (knitting, playing an instrument), and social activities (social clubs, going out for dinner). For activities with a component that crossed two categories, the primary function of the activity was reported.

## 3. Results

### 3.1. Participant Characteristics at 6-Month Follow-Up

A total of 2000 patients completed a 6-month follow-up questionnaire of post-stroke activity; 1549 (77%) were stroke survivors and 451(23%) TIA survivors (Table 1). Of those participants, 1045 (52%) provided a response to the question on barriers and 820 (41%) provided a response to the question on facilitators. Participant characteristics of respondents to the two questions were similar to all respondents at 6 months except forparticipants with no significant disability who were less likely to respond to the barriers question than overall respondents (17% less) and participants with a slight to moderate disability who were 12% more likely to respond to this question.

### 3.2. Barriers and Facilitators at 6 Month Follow-Up: Conceptual Themes

Twelve conceptual themes were identified, six each for barriers and facilitators. Barriers to returning to leisure activity were (1) physical difficulties (69%), (2) lower energy levels (17%), (3) loss of independence (11%), (4) psychological difficulties (10%), (5) hidden disabilities (7%), and delay in or lack of healthcare provision (3%). Facilitators were (1) family support (35%), (2) healthcare support (27%), (3) well-being and fitness (22%), (4) friendship support (20%), (5) self-management (19%), and (6) returning to normality (9%). A description of each theme (with supporting quotes) is provided in Table 2 and Table 3.

#### 3.2.1. Barriers: Distribution of Participant Characteristics

Barriers to activity were analysed by distribution of participant characteristic (Appendix A; Figure 1, Figure 2 and Figure 3). ‘*Physical health difficulties*’ was the greatest barrier experienced across all participant characteristics (Figure 1). Those participants with no significant disability reported this as a barrier the least (58%) and those with severe disability the most (81%). The second most reported barrier across the characteristics was ‘*lower energy levels*’ ranging from 8% for those with the most severe disability to 24% for those who were aged under 50. Exceptions to the rule were seen for those who did not experience post-stroke fatigue, where a ‘*loss of independence*’ (16%) was the second most reported barrier and for participants with a severe disability who identified a ‘*delay or lack of healthcare provision*’ more frequently. ‘*Psychological difficulties*’ and ‘*loss of independence*’ were the third most identified barriers across characteristics. *Psychological difficulties* were the most frequently reportedby adults <50 (18%), and *Loss of independence* without post-stroke fatigue (16%).

*Physical health difficulties* were by far the most common barrier reported across all living situations, types of transport use and employment status (Figure 2). The second most reported barrier across these categories was mostly *‘lower energy levels*’ except for those who live in care homes or hospital, where it was a *‘delay or lack of healthcare provision’and* for those who use public transport, walk oruse a bike for transport, it was ‘*loss of independence*’. For those who rely on a relative or friend for transport, both ‘*lower energy levels*’ and ‘*loss of independence*’ were identified equally. The third most common barriers reported were either ‘*loss of independence*’ or ‘*psychological difficulties*’. For those in full-time employment, ‘*psychological difficulties*’ were more commonly identified (21%) than ‘*loss of independence*’ (5%) whereas for those who were retired, the reverse order was true. 

#### 3.2.2. Barriers by Type of Important Activity

For all types of important activity, ‘*Physical difficulties*’ were prominently reported followed by ‘*lower energy levels*’ (Figure 3). The third most identified barriers across the activity groupings were ‘*loss of independence*’ for social activity, or ‘*psychological difficulties*’ for everyday and fitness & well-being activities. Those who participated in craft and hobby activities identified both barriers equally.

#### 3.2.3. Facilitators: Distribution of Participant Characteristics

Facilitators to activity were analysed by distribution of participant characteristic (Appendix A; Figure 4, Figure 5 and Figure 6). Overall, ‘*family support*’ was the most reported factor for facilitating a return to leisure activity across the participant characteristics (Figure 4). Those participants who did not experience post-stroke fatigue reported this theme the least (29%) and found ‘*well-being and fitness*’ more helpful (32%). Those participants with the most severe disability reported ‘family support’ the most (49%) but also found ‘*healthcare support*’ to be helpful (53%). *Healthcare support* was the second most identified facilitator across participant characteristics. There were some exceptions to this rule where ‘*well-being and fitness*’ was rated second most reported and this applied to those with TIA (23%), living in high socioeconomic deprivation (30%) or those with low disability (24%). For those under 50 years old (‘*friendship support*’) was the second most reported facilitator (38%), and for those without post-stroke fatigue it was ‘*family support*’ (29%). The third most identified facilitators were ‘*well-being and fitness*’ and ‘*friendship support*’. For participants who were either female, under 50 years old, in low socioeconomic deprivation or with depression, ‘*friendship support*’ was more commonly identified than ‘*well-being and fitness*’. Those participants who were below 50 years old or without post-stroke fatigue also identified ‘self-management’ as helpful. A desire to ‘*return to normality*’ was reported by a smaller number of participants and was not commonly identified for any specific characteristic.

Figure 5 illustrates the top three facilitating factors reported, based on participants’ living situation, transport use and employment status. *Family support* was the most frequently cited facilitator for most living situations, except for those living in care homes/hospital where it was ‘*healthcare support*’. *Family support* was also the most reported facilitator for the different categories of transport use and employment status, except for the categories of ‘walk/bike’ and ‘part-time’ worker, where it was ‘*healthcare support*’. The second most identified facilitator varied across categories. For living situation and those who live with a partner, it was ‘*healthcare support*’ and for those who live alone or with a relative/friend: ‘*friendship support*’. For those who live in care homes/hospital it was equally split between ‘*family*’ and ‘*friendship support*’. For transport use, those who relied on relatives/friends or public transport/taxi reported ‘*healthcare support*’ as the second most reported help but for those who drive their own car, it was ‘*well-being and fitness*’ and for those who walk or use a bike it was ‘*family support*’. Most categories of employment status reported ‘*healthcare support*’ as the second most identified theme except for part-time workers where it was ‘*family support*’. The third most identified facilitator for these categories was also varied. For living status and those who live alone it was ‘*healthcare support*’, for those who live with a partner ‘*wellbeing and fitness*’ and for those living with a relative or friend ‘*Self-management*’. For those living in care/hospital, ‘*well-being and fitness, self-management and returning to normality*’ facilitators were not reported by participants.

For transport use and those who drive their own car, ‘*self-management*’ was identified in the same proportions as ‘*healthcare support*’ for the third most identified theme. Those who relied on relatives or used public transport reported ‘friendship support’ as did those who walked or used a bike who equally reported self-management. The third most identified facilitator for employment status was ‘*well-being and fitness*’ except for participants in full-time employment where ‘*friendship support*’ (and ‘*self-management*’ were slightly more prevalent.

#### 3.2.4. Facilitators by Type of Important Activity

For all types of important activity, ‘*family support*’ was commonly identified as a facilitator (Figure 6). For those who enjoyed craft and hobby type activities ‘*friendship support*’ was slightly more reported. *Healthcare support* was also a notable facilitator for the different types of activity undertaken except for the craft and hobby group. Practising *well-being and fitness* as a facilitator to activity was identified for both the craft and hobby group and fitness and well-being groups. *Friendship support* was a notable facilitator for all groups except for fitness and well-being.

#### 3.2.5. Participants Who Were Older, Female or Living in an Area of High Socioeconomic Deprivation

In the prior study, [16] it was identified that those participants who were older, female and living in an area of high socioeconomic deprivation were more likely to experience a reduction in leisure engagement. This sequential analysis identified that the main barriers for these groups were ‘*physical difficulties*’ and *‘lower energy levels*’ (as for all participants) and either loss of independence (older participants) or psychological difficulties (females or living in high socioeconomic deprivation). The primary facilitator for these participants was the same as for all: family support. Healthcare support was the second most reported facilitator for those aged 70 and above, improving *‘well-being and fitness*’ for those living in an area of high deprivation and for females it was *‘friendship support*’.

## 4. Discussion

### 4.1. Findings

Our findings show that physical difficulties associated with stroke or TIA are a notable barrier to engagement in leisure activities at six months after the event, across all participant characteristics and activity types. As may be expected, the more severe the disability, the more notable this barrier becomes. Lower energy levels were also reported as a barrier for many participants and particularly for those who were younger, possibly due to having a greater impact on previous activity participation. For those with a severe disability, having less energy was not as much of as a hinderance, and receiving timely and appropriate healthcare provision was more important. Other barriers included loss of independence, psychological difficulties, hidden disability and lack of healthcare provision. Comparisons between stroke and TIA participants remained similar although *‘loss of independence*’ was more notable for those with stroke, as was *‘psychological difficulties*’ for TIA, a possible reflection of the physical impact of stroke compared to TIA. Analysis of barriers by living situation, transport use and employment status revealed similar findings to those overall, with some exceptions such as for participants who used public transport, where a loss of independence (perhaps from being unable to drive) was a greater barrier than lower energy levels. For facilitation, the support of family through both practical help and emotional encouragement, helped with activity engagement and most participants identified *‘family support*’ as the key facilitator. Those participants with the most severe disability relied on family support the most, although healthcare support was still the most reported facilitator for this group. The support of healthcare via rehabilitation therapies also remained important to many for improving ability and confidence to undertake activities, with the exception of younger, less tired and more able participants and those with TIA who were less reliant on this support. Other facilitators included well-being and fitness, friendship support, self-management and returning to normality. Analysis of facilitators by living situation, transport use, employment status and activity type followed a similar pattern to those overall. There were some exceptions, such as for those who worked part-time, where healthcare support was reported more often than family support, feasibly due to having a greater health need than full-time workers. No major differences were found for the groups of participants who had previously been identified as more at risk of reduced activity: females, older people and those living in an area of high socioeconomic deprivation. However, both females and those living in higher socioeconomic deprivation were more likely to describe *‘psychological difficulties*’ as a barrier than males and those living in lower socioeconomic deprivation. Females also identified *‘friendship support*’ as a facilitator more frequently than males and for those living in areas of higher deprivation, improving well-being and fitness was slightly more reported than healthcare support.

### 4.2. Comparison with Existing Literature and Guidelines

Physical limitations have previously been identified as a barrier to undertaking physical activity in older adults [21], and resuming previously valued activity post-stroke [22]. Conversely, the ability to walk a few hundred metres during the 2nd year post-stroke has been shown to have a positive effect on the frequency of social and leisure activities 10 years after stroke [23]. In this study, physical difficulties were often manifest alongside a decrease in energy levels that required more rest or sleep. Depletion of energy levels, lacking energy to undertake activity or requiring periods of rest within daily routines have similarly been reported elsewhere [22,24]. In our findings, a loss of independence also formed a practical and emotional barrier, particularly when the ability to drive was curtailed. Being dependent on others and being unable to plan and manage your own time and activities has previously emerged as a restricting condition on participation in social and leisure activity [15] and social isolation is often exacerbated by not being able to drive and get to places [24]. Other barriers identified elsewhere include psychological difficulties such as anxiety and lack/loss of confidence in social settings or the public sphere [21,22,24,25], hidden disability such as loss of confidence in one’s body and appearance [22] and feeling ‘cut off’ from healthcare support post-discharge or experiencing unmet rehabilitation needs leading to activity avoidance and social withdrawal [15,24].

Strong social support networks (family members and friends nearby, willing to help) have previously been acknowledged as helpful in resuming participation in social, leisure and everyday activity post-stroke [15,22,23,24,26]; Previous literature has also reported professional support, rehabilitation, provision of personal adapted equipment and information as helpful for pursuing new social and leisure activities [15,27]. In this study, practising ‘*well-being and fitness*’ such as eating well or undertaking exercise was a mechanism for improving health and ability, subsequently facilitating a return to activities, including those of a physical nature such as golf or the gym. Optimising physical and cognitive capacity, for example through exercise or skills training has been recognized before as a strategy for engagement in valued social and leisure activities post-stroke [28]. Other facilitators for engagement identified in this study have also been acknowledged previously such as self-management techniques including a positive attitude to stroke recovery, determination, persistence, self-belief and motivation to participate [15,22,24,28].

### 4.3. Clinical and Policy Implications

NICE clinical guidelines for stroke rehabilitation acknowledge the importance of engagement in activity post-stroke, and currently recommend that a comprehensive assessment on admission to hospital should take into account any activity limitations and participation restrictions [29]. Furthermore, they advise that people with stroke have goals set for their rehabilitation that focus on activity and participation. More guidance however would be beneficial on which members of the multi-disciplinary stroke team should complete these assessments and the use of validated tools for assessing activity and participation limitations.

To facilitate a return to leisure and social activity, whilst mitigating against recognised difficulties, and a readiness to engage, more is clearly required than healthcare provision alone. Where possible, a family-focussed approach to engagement could be encouraged during in-patient rehabilitation, exploring access to the support of family and friends. Options for facilitating such support could include an analysis of personal networks, and peer-led coaching or dialogue. A novel peer-led coaching intervention to improve post-stroke leisure and social participation in the community, was previously tested and reported a benefit to both coaches and stroke survivors at a personal level, for example by improving confidence [24]. Similarly, a mental-health model for peer-supported open dialogue, where family and friends attend healthcare meetings and help to make informed, collaborative decisions on organising care, received a positive endorsement from both NHS staff and patients [30]. An analysis of personal networks, including the function and quality of support provided, has also been reported to be helpful for exploring a patient’s social support in the context of chronic pain [31] and may also be useful for stroke survivors. It is worth noting however, that any such intervention in hospital would benefit from a co-ordinator who was tasked with identifying support networks, as co-ordinated multi-disciplinary team input has been shown to improve patient outcomes for stroke survivors [32].

Upon discharge from hospital, transition of care for all older adults could also include a focus on maintaining patients’ previous activities [25]. Once in the community setting, social prescribing may also offer practical support, including engagement in activities such as hobby clubs. This is not straightforward however for people with long-term neurological conditions who may experience issues with accessibility, adaptability, transport, psychological barriers such as reduced confidence or anxiety and concerns over acceptance by others [33]. Thus, these approaches require the involvement and investment of healthcare services.

It is also important to consider that service user preferences for support after stroke have indicated a desire for additional social and leisure activities that can be attended on their own and are provided by the hospital or community stroke team [34]. Approaches to acknowledge these preferences could even involve the use of an e-bike or e-trike, which despite barriers related to impairment have been reported as an opportunity to resume previously valued activity and facilitate social interaction post-stroke [35]. Alternatively, the use of tablet technology to support stroke recovery in older adults has been reported to be beneficial and easy to use, increasing participation in therapeutic and leisure activities and reducing boredom [36]. We must acknowledge however, that stroke-specific and age-related impairments can limit the use and functionality of information technology devices for people with stroke [37]. More importantly, for any intervention designed to promote engagement in social and leisure activities, we should also consider patients without existing ability and support in place. Lastly, it should be noted that re-engagement in social and leisure activities after stroke is a long-term process, subject to changes in life situation and health that may require various types of support during different stages of recovery and adaptation [28].

### 4.4. Strengths and Limitations

This study benefitted from a strong questionnaire response rate at baseline and follow-up (61%) and was unique in its analysis of qualitative findings by distribution of participant characteristic. This method enabled a more thorough understanding of specific issues and identified barriers and facilitators to engagement that were common to all participant groups. Building on the previous study’s findings of a 22% loss in activity post-stroke [16], this study helped to understand why this might happen and what might help to overcome this problem.

The study however would have benefitted from a more diverse sample to allow analysis based on ethnicity. Despite best efforts to recruit from geographically diverse sites, participants were predominantly white and an analysis by ethnicity was not possible. This is particularly relevant, as a higher proportion of people from black and ethnic groups have reported a negative change in number or type of leisure activities because of their stroke (around 80%) compared to people of white ethnicity (57%) [11]. The sample may also have been subject to selection bias due to the nature of response; completing and returning a questionnaire. Participants may only be representative of those who are able and willing to respond (the majority of respondents were more able) and are already engaged in leisure activity or would like to be. Furthermore, the open-text responses meant that there was no way of clarifying if no response meant no barriers or facilitators were identified or if they were just unreported.

### 4.5. Further Research

This study has raised further questions about how we support stroke survivors in returning to, or participating in, leisure activity post-stroke or TIA. Strong support networks may be key, but it is still unclear how to identify and utilise this support in practice, or support those who do not have family and friends available to help. Other areas for exploration include the impact of socioeconomic status on activity engagement. Individuals living in areas of high socioeconomic deprivation reported psychological difficulties more frequently than those in areas of low socioeconomic deprivation, yet financial constraints were not reported as a common barrier. More work needs to be undertaken to explain health inequity in this area.

## 5. Conclusions

Stroke and TIA survivors experience difficulties in returning to leisure activity post-stroke, predominantly related to physical capability and lack of energy. The support of family and friends, healthcare and well-being practices could help to overcome these issues through the provision of practical and emotional support. The rehabilitation processes of stroke and TIA patients may consider these support mechanisms when encouraging a return to activity post-stroke. More work needs to be undertaken in how these considerations could be implemented, and how to help patients who many not be able to access such support.

## Figures and Tables

**Figure 1 ijerph-19-14587-f001:**
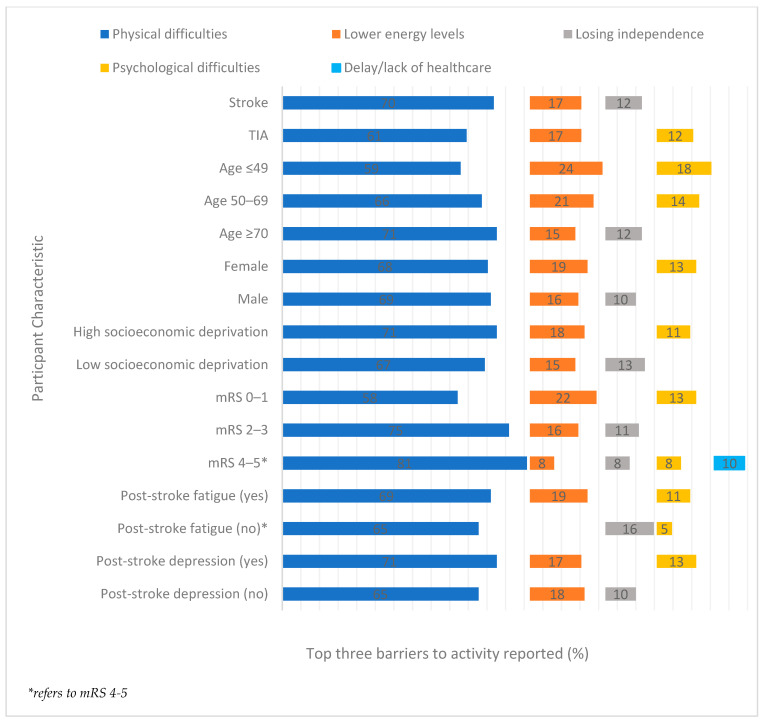
The three most reported barriers to activity (%) at 6 month follow-up by participant characteristic.

**Figure 2 ijerph-19-14587-f002:**
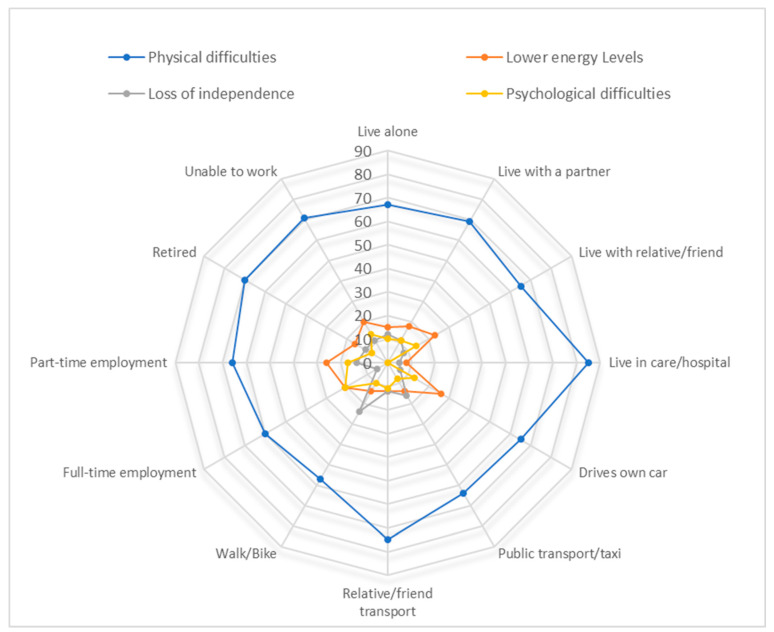
The three most reported barriers to activity (%) at 6 month follow-up by participant’s living situation, transport use and employment status.

**Figure 3 ijerph-19-14587-f003:**
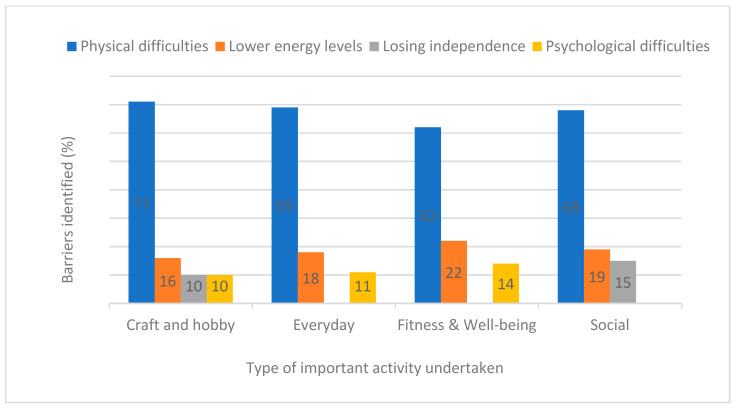
Top three reported barriers to activity(%) at 6 month follow-up by type of important activity undertaken.

**Figure 4 ijerph-19-14587-f004:**
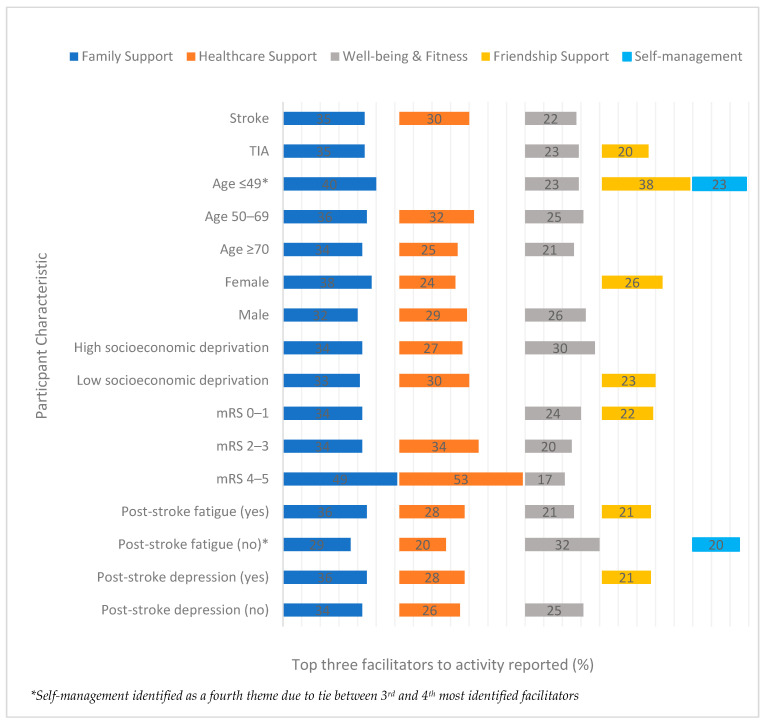
The three most reported facilitators to activity (%) at 6 month follow-up by participant characteristic.

**Figure 5 ijerph-19-14587-f005:**
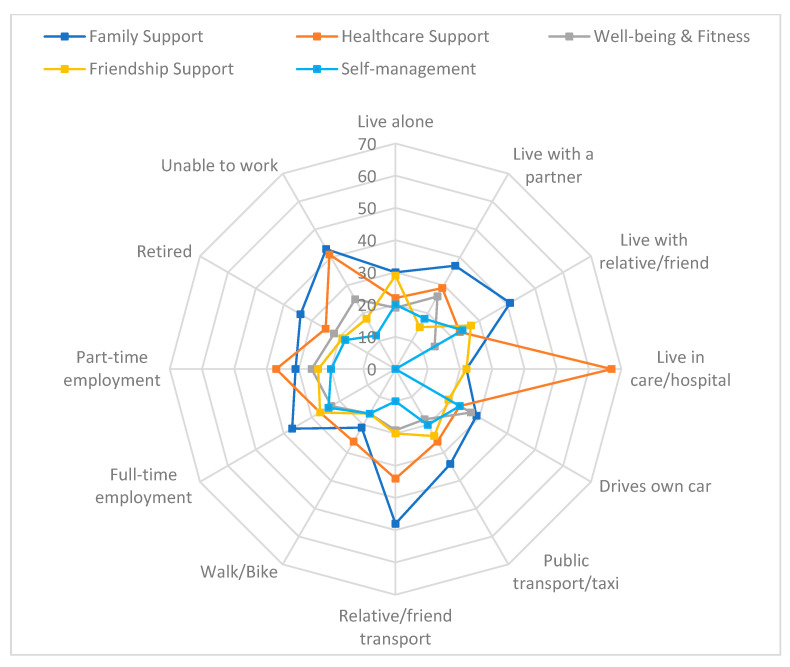
The three most reported facilitators to activity (%) at 6 month follow-up by participant’s living situation, transport use and employment status.

**Figure 6 ijerph-19-14587-f006:**
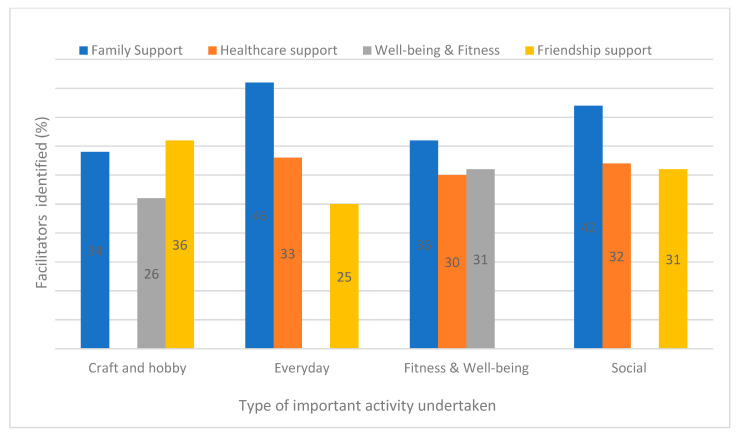
Top three reported facilitators to activity (%) at 6 month follow-up by type of important activity undertaken.

**Table 1 ijerph-19-14587-t001:** Participant characteristics for respondents to 6-month questionnaire and barrier/facilitator questions.

Participant Characteristics	Respondents to 6M Questionnaire n = 2000	Respondents to Barrier Question n = 1045	Respondents to Facilitator Question n = 820
Event type, n (%)			
Stroke	1549 (77.5)	872 (83.4)	673 (82.1)
TIA	451 (22.6)	173 (16.6)	147 (17.9)
Age, n (%)			
≤49	104 (5.2)	66 (6.3)	47 (5.7)
50–69	633 (31.7)	315 (30.2)	238 (29.0)
≥70	1263 (63.2)	664 (63.5)	535 (65.2)
Age, median (IQR)	73 (65–80)	74 (64–81)	73 (65–80)
Sex, n (%)			
Male	1167 (58.3)	569 (54.5)	474 (57.8)
Female	833 (41.7)	476 (45.6)	346 (42.2)
Ethnicity, n (%)			
White	1942 (97.1)	1013 (97.0)	794 (96.8)
Asian	24 (1.2)	16 (1.5)	10 (1.2)
Black	16 (0.8)	6 (0.6)	8 (1.0)
Mixed	7 (0.4)	4 (0.4)	2 (0.2)
Other	11 (0.6)	6 (0.6)	6 (0.7)
Socioeconomic Deprivation quintile, n (%) *			
1st *(High deprivation)*	286 (14.3)	160 (15.3)	99 (12.1)
2nd	353 (17.7)	186 (17.8)	147 (17.9)
3rd	445 (22.3)	234 (22.4)	171 (20.9)
4th	448 (22.4)	210 (20.1)	203 (24.8)
5th *(Low deprivation)*	467 (23.4)	255 (24.4)	200 (24.4)
modified Rankin Scale, n (%)			
0–1 *(no symptoms-no significant disability)*	1211 (60.6)	456 (43.6)	515 (62.8)
2–3 *(slight to moderate disability)*	569 (28.5)	423 (40.5)	239 (29.1)
4–5 *(severe disability)*	192 (9.6%)	147 (14.1)	59 (7.2)
Unknown	28 (1.4%)	19 (1.8)	7 (0.9)

* Data missing for 1 participant in response to 6-month questionnaire (n = 1999).

**Table 2 ijerph-19-14587-t002:** Conceptual themes identified for barriers to activity at 6 month follow-up, number of participants in each theme and proportion of response (%).

Theme *n* (%)	Description	Quotes
Physical difficulties 717 (69%)	Physical difficulties that restricted or curtailed the ability to undertake activity. This included mobility problems such as walking and balance, communication issues, and loss of fine movement for undertaking crafts or playing an instrument.	“I used to walk about 5 miles each day, now, although I am improving, I can’t walk far at all” “Gardening-lack of balance when bending” “Loss of fine motor movements in both hands make it difficult to write and impossible to do the crafts I enjoyed doing”“change of voice leading to not being able to enjoy amateur dramatics again”
Lower energy Levels 179 (17%)	A decline in energy levels described by participants as tiredness or fatigue, lack of stamina, needing to lie down or sleep and having to do things slowly.	“My hobby is cars and bikes but I get tired very easily then have to pack up” “My stamina is now significantly less, e.g., walking, in the garden, shopping, etc. My social-stamina is less because too much talking gives me a headache” “A sense of fatigue. Having to do things like gardening more slowly”
Loss of independence 111 (11%)	A loss of independence, in particular the ability to drive, inhibited engagement in activities away from the home. Alternative travel arrangements often required substantial planning and reliance on others. This restriction often led to low mood. The loss of driving as an enjoyable activity also had a negative impact.	“Not being able to drive makes it difficult to carry tools/plants down to my allotment. I wait until someone is available to give me a lift or I call a taxi” “Needing to make a plan for any activities and checking somebody can take me is very depressing” “I have been the family driver for fifty-eight years and have found it difficult to give up driving which I’ve always enjoyed”
Psychological difficulties 109 (10%)	Fear and anxiety were notable psychological difficulties including a fear of falling when outside, anxiety about having another stroke or feeling vulnerable from being out alone. Others reported a lack of confidence in resuming activity and some felt that it was other people who indicated uncertainty in their ability, resulting in feelings of frustration and anger. Feelings of low mood, depression and apathy were also often reported as a barrier to resuming activity.	“I am a boat owner and very keen angler but since my stroke I am very nervous about going to sea on my own” “I find that I have lost a lot of self -confidence in my game (golf). This has resulted in me being very upset when I play badly and affects my engagement and my readiness to be involved”. “I have been ‘petted’ on the head and shoulders by people, as if I am a bewildered child-this makes me want to stay home”
Hidden Disabilities 69 (7%)	Hidden disabilities such as a lack of concentration or maintaining focus were sometimes a barrier to undertaking everyday activities, hobbies and crafts. Memory problems and feeling overwhelmed by the environment affected confidence, especially for activities that required focus or socialising.	“Reading was a great pleasure but issues with sight and concentration mean it is now too difficult” “My memory is not as good as it was. This is so for general conversation. I avoid quiz games and find spelling more difficult” “Sensitivity to busy environments, noise and lights made things difficult*”*
Delay or lack of healthcare provision 31 (3%)	A lack of healthcare support such as physiotherapy provision and home adaptations typically delayed recovery and a return to activity (getting out of the house or care home).	“Lack of immediate support. It took over 12 weeks for physio to start and for adaptions and support to be arranged” “We still can’t get out due to no ramp/rails waiting to be put in place-very frustrating”

**Table 3 ijerph-19-14587-t003:** Conceptual themes identified for facilitators to activity at 6 month follow-up, number of participants in each theme and proportion of response (%).

Theme *n* (% of Response)	Description	Quotes
Family Support 286 (35%)	Family support was both emotional and practical in nature. Encouragement and reassurance from family members helped to build confidence in returning to activities, from lunches out with family, to more organised activities such as attending a social club. Family support also included practical help with transport, mobility and facilitating a return to activities in addition to help with chores, hospital visits, etc.	“Partner has helped in giving me encouragement to return to singing with the choir” “Family support has been vital to resuming my favourite activities” “My daughter takes me out as much as possible and arranged art club bingo and takes me swimming/shopping and helps me with everything”
Healthcare Support 221 (27%)	Provision of rehabilitation therapies offered from healthcare services such as physiotherapy, occupational therapy and speech therapy were thought to be beneficial in improving ability such as balance and movement and subsequently enabling engagement in activity. Therapists also helped to install confidence that patients would return to activities in time. Stroke services including follow-ups, early supported discharge teams and the Stroke Association were referred to as informative, helpful and encouraging. Provision of mobility aids, equipment and appropriate medication were also reported as beneficial.	“Physios/OTs helped a lot from Stroke Team when I was discharged from hospital. Huge help to get me walking with my dog again (alone). Gave me more confidence” “I am sure with my physio’s help I will get back to my clubs and coffees/lunches out with friends” “All of the help and support of the stroke team has helped me with exercises and food a great deal”
Well-being and fitness 183 (22%)	Improving well-being was a means to re-engaging in activities and was practised by eating well, giving up smoking, losing weight and achieving a better work/life balance. Needing more rest was often necessary and was achieved through sitting breaks, sleeps during the day, regular bedtime, and more relaxation. Exercise and keeping active helped to improve general fitness for other activities, and walking was a popular choice. Many participants also enjoyed returning to previously enjoyed activities such as the gym, yoga or golf.	“Sleeping more and keeping the nutrition good” “Wanting to get fit and keeping healthy, by going back into the gym now, I feel a lot better in my mind, and health” “Walking also benefits my balance particularly when I’m playing the guitar and singing with a band”
Friendship Support 164 (20%)	As with family support, friends provided encouragement and practical help in returning to social activity. In addition, the continuation of previous social patterns and maintaining friendship was valued and provided further encouragement to move forward with recovery.	“My friends at my sheltered dwelling have encouraged me to help out making afternoon tea for our regular get togethers” “Friends in a fishing club have helped me fishing by carrying fishing tackle etc and helping me setting up if I need” “The fact that these activities are part of my social life where I meet friends and that continuing as before, has helped me move on”
Self-management 152 (19%)	Self-management techniques such as taking things slowly, setting goals and acceptance of limitations helped to build confidence. Self-belief, determination and a positive attitude were also beneficial in getting through difficulties and not giving up. Having the motivation to do activities again for pleasure, fitness or wanting to be part of life also helped to build resilience, adapt to changes and overcome difficulties.	“Taking things slowly and not thinking it will be just as it was. Enjoying what you can do, not what you can’t do” “I am also very independent and strong-willed, so very determined to get on as well as possible” “Positive mental attitude and a ‘can do’ approach-adapt and overcome”
Returning to normality 70 (9%)	The desire to get back to normal or return to old routines was a motivator for resuming activity. Returning to previous routines also offered a sense of reassurance that pre-stroke activities were possible. Time and patience were mentioned as a means of achieving the goal of returning to normal. Managing change helped to form a new normality.	“Getting back to normal, doing the things I like doing” “I think it’s putting everything behind me and focussing on having a normal life as before’ ‘Managing change and finding new ways of thinking and doing”

## Data Availability

The data presented in this study are available on request from the corresponding author.

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
