# Peer review of "Returning to Leisure Activity Post-Stroke: Barriers and Facilitators to Engagement"

_ijerph, 2022, doi:10.3390/ijerph192114587_

Round 1

Reviewer 1 Report

Harrison et al present data resulting from questionnaire on barriers and facilitators to return to leisure activities after stroke. This work benefits from a large sample size and a high response rate. The results are not surprising, however they describe well the situation. I do not have major concerns.

All necessary information is presented, the results might be a bit too detailed at some point. The text and tables are highly redundant. Being more concise while pointing out the important points might help the reader. For example 3.1 could be shortened. Furthermore, in line 188-189 the authors mentioned no differences between the groups while in the next line they point out the only relevant difference.

Figures: Is there a reason why different types of graphs were used for similar data analysis (figures 2 and 5 vs figures 3 and 6)?

The legend in figure 2 is not complete.

Author Response

Thank you for your positive and useful comments.  Please find my reply below:

'the results might be a bit too detailed at some point.  The text and tables are highly redundant.  being more concise while pointing out the important points might help the reader'

-results have been made more concise and clear.  For example, 3.1 has been reduced and the text describing analysis by participant characteristics has been made clearer. 

Anything that is potentially confusing in the results has been amended including the combined proportion ranges identified for barriers and facilitators by living situation, employment status and transport use (e.g. 8-34% etc.).  Figures 2 and 4 now focus on the top three themes identified (like the other figures) and the describing text talks through each theme rather than going through each category of living situation etc.  This matches these sections to the rest of the paper making it more logical to read and understand.

'is there a reason why different types of graphs were used' -just for presenting purposes as figures 2 and 4 needed to present 3 different groups of individual characteristics and the spider diagram helped to reflect this.

Reviewer 2 Report

The issue raised by the authors is of some importance. The mixed research method is used to study the obstacles and promoting factors of stroke or TIA patients' recovery or participation in leisure activities, which has certain innovation and practical significance. However, There are still some places that need to be revised, and I hope the author will revise them seriously.

1. 1. It is suggested to adjust the position.4.2. Strengths and limitations put after  4.5. 

2.Pay attention to the format of references.

3. The results of the paper have to be adjusted greatly. family support and friendship support can be merged.

loss of independence hidden disalibility and physical difficulties the author considered if there are can be merged. I think there are many repetitions.

4. There is something wrong with the presentation form of Figure 1 . It is suggested to modify it. Each distribution can be connected together, but there is a difference in color. In addition, there are many kinds of ordinate, but the division is not clear.

5. Some of the results don't see where the changes are. That is, the changes of recreational activities after baseline and follow-up for six months.

6. Part of the discussion lacks pertinence, and the research results are not discussed in depth.

Author Response

thank you for your useful comments.  Please find my response below:

1.1 it is suggested to adjust the position.  I wasn't sure what this refers to?

4.2. Strengths and limitations put after 4.5. Have amended.

2.Pay attention to the format of references.  Again, not sure which formatting issues are being referred to?

3. The results of the paper have to be adjusted greatly. ①family support and friendship support can be merged, loss of independence 、hidden disalibility and physical difficulties, the author considered if there are can be merged. I think there are many repetitions.  -Unfortunately, we are unable to merge these categories as we would lose the detail, e.g. 'loss of independence' has specific reference to loss of driving and therefore the ability to attend activity.  This is quite different from loss of physical ability meaning someone can no longer play golf for example.  Also, all the participant analysis has been undertaken on these groupings and it would require a major revision to re-calculate the proportions of barriers and facilitators identified for new groupings.  We have however, simplified the results section, including avoidance of repetition, clearer language and greater consistency between text and tables.

4. There is something wrong with the presentation form of Figure 1 . It is suggested to modify it. Each distribution can be connected together, but there is a difference in color. In addition, there are many kinds of ordinate, but the division is not clear.

-We have added gridlines to try and emphasise the difference between columns, the different colours represent different themes identified.  The x axis label has also been amended to improve understanding, e.g. 'top three barriers to activity reported (%)'

5. Some of the results don't see where the changes are. That is, the changes of recreational activities after baseline and follow-up for six months. -have added 6 month follow-up to charts and headings to emphasise the time period

6. Part of the discussion lacks pertinence, and the research results are not discussed in depth.  -Have added more discussion about the results and their potential meaning.

Reviewer 3 Report

This is a very nice study that explored factors related to leisure activity post-stroke, which is very important to promote participation and quality of life in this population. Though qualitative research often devalued as less scientific than quantitative research, it seems to give more strength to this study by identifying barriers and facilitators from the participants' own statements. As a result, this study not only encompasses previously known factors but also provides comprehensive understanding to help develop future policy.

Please check * in figure 1 marked after post-stroke fatigue(no). I cannot find its explanation. It may be inserted by mistake.

Thank you for your contribution.

Author Response

Thank you for your positive and useful comments.  Please find my response below:

Please check * in figure 1 marked after post-stroke fatigue(no). I cannot find its explanation. It may be inserted by mistake.  The statement explaining the asterix* was below figure 1 but I've moved it up now to within the figure.